# Production of Bioactive Compounds in *Grammatophyllum speciosum* Blume Using Bioreactor Cultures Under Elicitation with Sodium Chloride

**DOI:** 10.3390/plants14193083

**Published:** 2025-10-06

**Authors:** Jittraporn Chusrisom, Gadewara Matmarurat, Nattanan Panjaworayan T-Thienprasert, Wannarat Phonphoem, Pattama Tongkok

**Affiliations:** 1Kasetsart Agricultural and Agro-Industrial Product Improvement Institute, Kasetsart University, Bangkok 10900, Thailand; jittraporn.chus@ku.th (J.C.); gadewara.mat@ku.th (G.M.); 2Department of Biochemistry, Faculty of Science, Kasetsart University, Bangkok 10900, Thailand; fscinnp@ku.ac.th (N.P.T.-T.); wannarat.p@ku.ac.th (W.P.); 3Department of Fishery Biology, Faculty of Fisheries, Kasetsart University, Bangkok 10900, Thailand

**Keywords:** antioxidant activity, biological activity, bioactive compounds, medicinal plant, micropropagation, temporary immersion bioreactors

## Abstract

*Grammatophyllum speciosum* Blume is an endangered wild orchid with medicinal properties. In this research, we propagated *G. speciosum* from vegetative organs grown under aseptic conditions. Subsequently, salinity stress was applied at the plantlet stage to investigate its effect on the accumulation of bioactive compounds. Half-strength Murashige and Skoog (½ MS) medium supplemented with a combination of 1 mg of L^−1^ 1-naphthaleneacetic acid (NAA) and 0.5 mg of L^−1^ 6-benzylaminopurine (BAP) proved to be a more suitable medium for shoot formation (32.33 ± 2.52 shoots per explant). The protocorm-like bodies, derived from embryogenic callus, were transferred into a temporary immersion bioreactor (TIB) system; 10-min of immersion every 3 h enhanced the maximum number of shoots, shoot height, and the fresh growth index (127.00 ± 2.16, 5.00 ± 0.51 cm and 4.26 ± 0.52, respectively). The proliferated plantlets from the TIB system successfully rooted in Vacin and Went medium. Furthermore, the plantlets were maintained in ½ MS medium supplemented with sodium chloride (NaCl) (0, 50, 100 or 200 µM) under a white light-emitting diode for 72 h to determine the total phenolic content (TPC) in the in vitro cultures. The TPC was highest in the medium with 100 µM of NaCl (111.06 ± 2.24 mg gallic acid equivalent g^−1^ dry weight), the diphenyl picrylhydrazyl antioxidant activity was 24.50 ± 0.76% and ferric-reducing antioxidant power values were in the range 2441.79 ± 1.21 to 2491.96 ± 3.23 µM ascorbic acid equivalent g^−1^ dry weight. The *G. speciosum* extracts showed antibacterial activity against acne pathogens, with minimum inhibitory concentration and minimum bactericidal concentration values in the ranges 6.4–12.8 mg mL^−1^ and 12.8–25.6 mg mL^−1^, respectively.

## 1. Introduction

*Grammatophyllum speciosum* Blume is an endangered wild orchid with various medicinal properties and is distributeds across the southern islands of the Pacific Ocean and the tropical regions of Southeast Asia, including Thailand [1]. It is commonly known as the Giant orchid or Tiger orchid, and in Thailand, it is known as ‘waan phet cha hueng’. *G. speciosum* is used in Thai traditional medicine for the treatment of fever, insect bites, and skin rashes and as an anti-inflammatory remedy [2]. It is currently known that *G. speciosum* pseudobulb ethanolic extract exhibits various promising pharmacological activities for the treatment of breast cancer [3]. In addition, this extract has been utilized in the cosmetics industry for its antioxidant, anti-aging, and anti-collagenase activities [4], as well as for its skin-whitening efficacy [5]. These therapeutic effects result from the biological compounds produced through the plant’s metabolism. Previous studies reported that *G. speciosum* contains flavones, alkaloids, a sesquiterpenoid and a stilbenoid, which have demonstrated the ability to reduce the viability of human cancer cells (e.g., A375), highlighting the pharmacological potential of this species [6].

The cultivation of plant cells, tissues, and organs is crucial for investigating the processes associated with plant growth and development [7,8]. Such cultivation produces disease-free plants (plants free from known pathogens) that are used for breeding, propagation, and the conservation of endangered plant species. In vitro plant propagation presents a viable alternative for bioactive compound production [9]. Protocorm-like bodies (PLBs) are embryo-like structures derived from orchid explants in tissue culture. They can develop into genetically identical plantlets through somatic embryogenesis, providing an efficient method for clonal propagation [10]. A temporary immersion bioreactor (TIB) is an innovative system designed for plant tissue culture using a liquid medium. This system excels in simultaneously and efficiently producing large quantities of plants by providing a suitable medium and controlling the timing of nutrient delivery. An essential function of the TIB system is ensuring that plant tissues are not continuously submerged in the liquid medium, which is especially valuable in preventing them from becoming hyperhydric [11]. The first TIB system for banana propagation was reported by Alvard [12]. Subsequently, this system has been used globally for the large-scale micropropagation of crops [13,14,15,16], medicinal plants [17,18], and other economic plants [13,19]. The shoot multiplication rate can be significantly enhanced from two-fold to more than ten-fold in a TIB, compared to that obtained through semi-solid culture methods [20,21,22]. In addition, micropropagation using TIB systems has been reported for several orchid species, such as *Cattleya tigrine* [23], *Dendrobium* sp. [24,25], *Phalaenopsis* sp. [26], *Vanilla planifolia* [27], and *Vanda tricolor* [11].

Therefore, the use of a TIB could be an interesting approach to cultivating *G. speciosum* because this plant reproduces via seed propagation or the separation of pups (suckers). The conventional seeding method lacks effectiveness, as seeds tend to disperse naturally, resulting in a low survival rate due to insufficient nutrients for germination [28]. In addition, the propagation via suckers is slow and requires a long period for mother plant establishment [29]. Importantly, the propagation of *G. speciosum* using aseptic techniques is highly advantageous because it enables the rapid production of high-quality plants within a short timeframe. This not only ensures a sufficient supply for immediate use but also facilitates the conservation and preservation of the mother plant for future applications and sustainability. Consequently, in the current research, we evaluated the efficiency of cultivating *G. speciosum* from vegetative organs grown in various media for shoot and root inductions. In addition, we determined suitable culture conditions that enhanced bioactive compound production under NaCl stress and the optimal immersion time in the TIB system to efficiently stimulate shoot multiplication. Furthermore, we examined the biological properties of *G. speciosum* crude extracts.

Protocorm-like bodies (PLBs) derived from *G. speciosum* explants under aseptic conditions are hypothesized to be efficiently propagated using a temporary immersion bioreactor (TIB) system. Plantlets obtained from this propagation were subsequently used in separate NaCl stress experiments to investigate the accumulation of bioactive compounds. This study represents the first report establishing a TIB-based micropropagation protocol for *G. speciosum*.

## 2. Results and Discussion

### 2.1. Induction of Protocorm-like Bodies

The shoot apexes and nodal segments of *Grammatophyllum speciosum* Blume cultivated in various media induced PLBs (Figure 1a), which visibly appeared after 21 days of culture. However, the explants of *G. speciosum* showed different growth and development. Some explants formed embryogenic calluses that were clearly visible on their nodes after 14 days of culture. These embryogenic calluses covered the node surface and varied from white to milky white (Figure 1b). Shoot formation was clearly detected on the surface of the shoot apexes and showed good growth after 28 days of culture. The highest number of these PLBs was observed at 35 days and remained consistent thereafter until 42 days. The PLBs displayed vigorous growth and demonstrated the capacity to develop into mature shoots (Figure 1c–j).

The highest number of shoots due to the induction of PLBs was observed in the T2 medium compared to the other media (Figure 2 and Table 1). Consequently, T2 was adopted as the suitable medium for shoot formation (35.00 ± 3.00). The sample with the highest number of shoots had the highest fresh growth index (9.92 ± 1.31) after 42 days. The shoot numbers of T3, T4, T5 and T6 showed 18.67 ± 2.08 12.33 ± 1.53 13.67 ± 1.53 6.67 ± 0.58, respectively. Meanwhile, the T3 media formed similar shoot numbers and fresh growth index values to that of T1 (control). However, the T4, and T5 media showed similar fresh growth index values to that of T1 but had different shoot numbers, and the shoots appeared to be fully tillering and had large stems (Figure 2). Samala [30] investigated the PLB formation of *G. speciosum* in a conventional solid medium (½ MS + BA + NAA), reporting an increase in the number of PLBs to 6.87 shoots explant^−1^ after 70 days of cultivation. Notably, in the current work the formation of PLBs was successfully induced in a liquid medium (½ MS + 1 mg L^−1^ BAP + 0.5 mg L^−1^ NAA), resulting in 35.00 ± 3.00 shoots per explant within 42 days (Table 1).

The formation of PLBs in liquid medium within a shorter cultivation time was compared to that in solid medium. Among the concentrations evaluated, the BAP and NAA combination promoted the highest shoot production in *G. speciosum*. The interaction between auxin and cytokinin regulates meristem development, which is important in creating the whole plant body [31]. The presence of elevated levels of cytokinins such as BAP influences apical dominance by promoting the formation of axillary and adventitious shoots; conversely, higher levels of auxin have been found to activate the formation of adventitious roots [32,33]. Other studies have demonstrated that optimal shoot proliferation in different cultivars is achieved by manipulating the cytokinin-to-auxin ratio. In a study conducted by Madhulatha [34], a pulse treatment involving a combination of 6-benzylaminopurine (BAP) and kinetin in a 1:1 ratio increased shoot formation in banana (*Musa* spp.). Jafari [35] revealed that a BAP concentration of 33 μM stimulated the multiplication of abnormal shoots in banana (*Musa acuminata* cv. Berangan), while shoot elongation was supported by indole-3-acetic acid. Additionally, Rahman [36] reported that the combination of 4.0 mg L^−1^ of BAP and 1.5 g L^−1^ of NAA activated shoot formation in banana (*Musa sapientum*). It the current study, it was shown that ½ MS medium supplemented with 1 mg L^−1^ of BAP and 0.5 mg L^−1^ of NAA was appropriate for shoot multiplication in *G. speciosum.*

### 2.2. Effect of Different Immersion Times and Frequencies on Shoot Multiplication

In the TIB system, combinations of immersion time and frequency were observed to affect shoot multiplication efficiency of *Grammatophyllum speciosum* Blume (Figure 3). Notably, there were significant differences in growth parameters between immersion times of 5 min and 10 min. Immersion for 10 min every 3 h (IF4) enhanced the maximum number of shoots per explant and the maximum shoot height, as well as the fresh growth index (127.00 ± 2.16 shoots per explant, 5.00 ± 0.51 cm, and 4.26 ± 0.52, respectively). As shown in Table 2, the growth parameters after 10 min immersion decreased significantly when the immersion intervals reached 6 h (IF5) and 12 h (IF6), respectively. With immersion for 5 min, there were no observable differences in the number of shoots per explant or shoot height between 3, 6, and 12 h intervals (IF1, IF2, and IF3, respectively). Nevertheless, it is worth noting that the immersion for 5 min every 3 h (IF1) resulted in the lowest number of shoots per explant and the lowest fresh growth index compared to those in all other immersion treatments. The frequency of immersion affects various processes, including stomatal function, nutrient absorption, and the occurrence of hyperhydration such as translucent or glassy tissues, thickened stems, and elongated, twisted, and brittle leaves [37]. Moreover, previous studies revealed that low immersion frequencies limit shoot multiplication because of reduced nutrient availability, whereas high immersion frequencies promoted shoot formation but also induced hyperhydric shoot development [15,38,39]. In this study, multiple shoots of all immersion frequencies did not exhibit hyperhydric symptoms. The number of shoots increased under higher immersion frequencies with an immersion duration of 10 min. Extended immersion periods also enhanced shoot propagation and growth, likely due to the prolonged opportunity for nutrient absorption by the explants. Therefore, the optimal system for growth and development to increase the number of shoots of *G. speciosum* was immersion for 10 min every 3 h (IF4).

### 2.3. Root Induction

The multiple *Grammatophyllum speciosum* Blume shoots cultured in the TIB system were subsequently transferred to six different rooting media to induce roots (Figure 4), all of which supported the growth of *G. speciosum* (Figure 5). Notably, successful root formation occurred in the RT1, RT2, RT5, and RT6 media. The RT5 medium promoted the highest root number and longest root length (Figure 1k and Figure 5 and Table 3). Conversely, root formation was not observed in the RT3 and RT4 media, both of which were based on the full MS medium. Additionally, the plantlets exhibited atypical growth characterized by abnormal elongated stems and unhealthy plantlet, indicating possible symptoms of excessive nutrient exposure in the *G. speciosum* culture. The composition of the MS medium consisted of both macro- and micro-nutrients [40]. On the other hand, the VW medium comprised three macronutrients (potassium nitrate, ammonium sulfate, and monopotassium phosphate) along with essential micronutrients (magnesium sulfate, manganese sulfate, and ferric sulfate), according to the report by Vacin and Went [41]. The current results are consistent with their findings, which indicate that the VW medium provided suitable nutrients for promoting the root growth of *G. speciosum* (Figure 5d,e). Therefore, nutrient uptake could potentially play a significant role in physiology [42] and morphological development. In addition, the RT5 medium contained coconut water and bananas; another study highlighted the growth-enhancing effects of coconut water on orchids, leading to elevated propagation rates [43]. Furthermore, the combined application of coconut water and banana additives reportedly enhanced root regeneration in Florida’s native orchid species [44]. Therefore, supplementary fresh coconut water and banana in the VW medium played an important role in root induction, without negative effects on the growth of *G. speciosum* shoot. Although the current study focused on root growth using different media (MS, ½ MS, VW), further studies using identical growth regulator and salt combinations across all media could provide additional insights into optimizing both growth development and subsequent bioactive compound production.

### 2.4. Effect of NaCl Stress on Total Phenolic Content

The total phenolic content of the plantlets extracted with ethyl alcohol was determined using the regression equation determined for gallic acid (y = 0.0034x + 0.0005; R^2^ = 0.999). The highest total phenolic content was found in the 100 µM NaCl treatment (111.06 ± 2.24 mg GAE g^−1^ DW) compared to the 0 µM NaCl treatment (control = 83.71 ± 2.47 mg GAE g^−1^ DW), as shown in Figure 6a. Meanwhile, the total phenolic contents were 92.39 ± 2.86 and 82.50 ± 2.63 mg GAE g^−1^ DW in the 50 and 200 µM NaCl treatments, respectively. Chowjarean [45] reported that analyzed phenolic compounds from naturally grown *Grammatophyllum speciosum* Blume ethanolic extract had a total phenolic content of 48.19 ± 0.39 mg EGCG equivalent g^−1^ extract. Cultivating *G. speciosum* using aseptic techniques can significantly reduce the required growth period to produce high-quality biological extracts. Importantly, this method does not reduce the total quantity of phenolic compounds. Furthermore, this approach has the potential to regulate the increase in the total phenolic content by incorporating nutrients (NaCl) into the cultivation medium. This aligns with research by Nag & Kumaria [46], which indicated that 100 µM NaCl can stimulate the production of phenolic compounds in *Vanda coerulea* Griff. ex Lindl. Similarly, the current findings showed a significant increase in the total phenolic content after NaCl treatment, which is responsible for protecting plants from the harmful influences of salt stress [47,48,49,50]. Dendrobium orchids, respond to salinity by performing osmotic adjustment and sequestering Na^+^ or Cl^−^ in the roots, preventing excessive ion accumulation in the shoots. This suggests that the applied NaCl concentration was within a physiologically tolerable range, allowing enhanced phenolic production without severely compromising growth [51]. However, notably, exposure to high salinity can lead to an ion imbalance, hyper-osmotic stress, and subsequent osmotic and oxidative stresses in some plants [52]. In addition, salinity stress stimulates the phenylpropanoid biosynthetic pathway; accordingly, it supports phenolic compound accumulation [53]. Mechanistically, salinity stress is known to induce oxidative stress in several plant cell, leading to the generation of reactive oxygen species (ROS) [54,55]. Plants have evolved specific mechanisms to detoxify ROS, which include the activation of antioxidant enzymes such as superoxide dismutase, catalase, and peroxidase as well as non-enzymatic antioxidants including phenolic compounds, alkaloids and ascorbic acid [56,57,58]. Moreover, the accumulation of reactive oxygen species (ROS) under salt stress can act as a signaling molecule, triggering the activation of specific transcription factors (TFs) and differentially expressed genes (DEGs) [59,60]. These transcription factors, such as MYB, WRKY, AP2/ERF-ERF, NAC, and bZIP regulate the expression of enzymes in the phenylpropanoid biosynthetic pathway [61,62,63], leading to enhanced synthesis and accumulation of phenolic compounds, which function as antioxidants to mitigate oxidative damage. According to these reports, within certain physiological ranges, salinity can act as an elicitor that promotes the accumulation of phenolic compounds in plants. Environmental stress combinations are an interesting subject and have led to high levels of secondary metabolite production in *G. speciosum*.

### 2.5. Antioxidant Activity

The antioxidant activity of *Grammatophyllum speciosum* Blume extract based on DPPH and FRAP assays is shown in Figure 6. The extracts treated with 100 µM of NaCl showed the highest potential DPPH scavenging activity compared with those under the untreated condition (0 µM NaCl) (24.50 ± 0.76% and 3.93 ± 0.36%, respectively), as shown in Figure 6b. The DPPH scavenging activity levels in extracts under the 50 and 200 µM NaCl treatments were 14.59 ± 1.04% and 16.29 ± 0.84%, respectively. Previously, the percentage of DPPH scavenging activity in *Cottonia peduncularis* and *Dendrobium* spp. orchids were found to be 29.83% and 32.58%, respectively [64,65]. The FRAP assay is based on the capacity of antioxidants to reduce [Fe(III)(TPTZ)_2_]^3+^ complexes to [Fe(II)(TPTZ)_2_]^2+^, and the FRAP value is calculated using a standard curve of ascorbic acid concentrations ranging from 0 to 1000 µM (y = 0.0012x + 0.0022; R^2^ = 0.999). The *G. speciosum* extracts treated with 100 µM of NaCl had reducing activity within the range 2441.79 ± 1.21 to 2491.96 ± 3.23 µM AAE g^−1^ DW (Figure 6c), and the FRAP values were within the range 1129.48 ± 0.57 to 1132.62 ± 1.00 for the control (0 µM NaCl). Meanwhile, the *G. speciosum* extracts treated with 50 and 200 µM NaCl had reducing activity within the range 1904.17 ± 3.09 to 1927.74 ± 2.69 µM AAE g^−1^ DW and 2385.85 ± 3.35 to 2397.88 ± 2.68 µM AAE g^−1^ DW, respectively. In addition, these results present a positive correlation between the total phenolic content and the antioxidant capacity in the *G. speciosum* plantlets treated with 100 µM of NaCl. Elsewhere, the radical scavenging activity in *Luisia zeylanica* [66] and Nepal wild orchids [67] produced antioxidant activity levels that were correlated positively with total polyphenolics. The findings imply that exposure to salt stress could contribute to enhanced antioxidant activity in the plantlets of *G. speciosum* under the TIB system.

### 2.6. Antibacterial Activity

According to the above results, the highest total phenolic content was observed in the *Grammatophyllum speciosum* Blume plantlets treated with 100 µM NaCl. Consequently, they were selected for antibacterial activity evaluation. The *G. speciosum* crude extracts showed antibacterial action at the minimum inhibitory concentration (MIC) and minimum bactericidal concentration (MBC) values (Table 4). Different bacteria were inhibited and eliminated at different concentrations of the extract. *Propionibacterium acnes* (6.4 mg mL^−1^) had the lowest MIC, followed by *Staphylococcus aureus* (12.8 mg mL^−1^) and *Staphylococcus epidermidis* (25.6 mg mL^−1^). Each bacteria species died when it was treated with an extract of twice the MIC, as seen in the MBCs in Table 4. Other studies reported that the bioactive compounds isolated from *G. speciosum* extract included gastrodin [45,68] and vitexin and orientin [4]. Gastrodin is a phenolic glycoside [69], whereas vitexin and orientin are flavonoid derivatives [4] belonging to a group of phenolic compounds that can be found in plants. Polyphenols exhibit antimicrobial activity through interactions with bacterial cell surfaces and the disruption of membrane integrity [70,71]. González-Cortazar [72] reported that gastrodin was active against *Staphylococcus aureus* (MIC > 0.1 mg mL^−1^). Likewise, the MIC values of vitexin (>1.0 mg mL^−1^) and orientin (0.5 mg mL^−1^) showed that they have antibacterial activity against *Staphylococcus aureus*; however, the mechanism of action is still unknown [73]. In addition, other orchid extracts have exhibited activity against acne bacteria. Irimescu [74] and Olivares [75] reported that extracts of different parts of *Phalaenopsis* orchids (roots, leaves, and stems) had inhibitory activity against different bacteria. They found that the stem and root extracts could be antimicrobial agents for *Staphylococcus aureus* and *Staphylococcus epidermidis*, respectively [74]. Additionally, a mixture of root and leaf extracts showed antimicrobial activity against *Propionibacterium acnes* [75]. This inhibitory activity could be linked to the presence of bioactive compounds in orchid extracts, as the occurrence of phenolic compounds in *G. speciosum* plantlets revealed potential antimicrobial activity against pathogens causing acne.

## 3. Materials and Methods

### 3.1. Plant Materials and Induction of Protocorm-like Bodies

The shoot apexes and nodal segment of *Grammatophyllum speciosum* Blume were cultured in half-strength Murashige and Skoog (½ MS) liquid medium [40] supplemented with 20 g L^−1^ of sucrose and various concentrations of 6-benzylaminopurine (BAP) (Sigma-Aldrich, St. Louis, MO, USA) at 0.5, 1.0 or 2.0 mg L^−1^ and 1-naphthaleneacetic acid (NAA) (Sigma-Aldrich, St. Louis, MO, USA) at 1.0 or 2.0 mg L^−1^.

The pH of each liquid medium was adjusted to 5.7 and then, 20 mL of each medium was dispersed in a 250 mL tissue culture bottle. Subsequently, all media were sterilized using an autoclave (Zealway, Xiamen, China) at 121 °C and 105 kPa of pressure for 20 min. The cultures were shaken at 110 rotations per minute, 25 ± 1 °C, and a humidity of 51 ± 2% under a white light-emitting diode (LED) at an intensity of 37 µmol m^−2^ s^−1^ for 16 h day^−1^ for 42 days.

### 3.2. Effect of Immersion Time and Frequency on Shoot Multiplication Efficiency

A sample of 0.5 cm protocorm-like bodies (PLBs) was inoculated into a 720 mL glass chamber of a temporary immersion bioreactor (TIB). Another chamber contained 100 mL of ½ MS liquid medium supplemented with 20 g L^−1^ of sucrose, 1 mg L^−1^ of 6-benzylaminopurine (BAP), and 0.5 mg L^−1^ of 1-naphthaleneacetic acid (NAA), as shown in Figure 7. The growth chamber was equipped with an air compressor and a controller box programmed to regulate the frequency and duration of medium supply (Figure 8). The TIB equipment was custom-built in Plant Tissue Culture Laboratory, Kasetsart Agricultural and Agro-Industrial Product Improvement Institute, Kasetsart University, Thailand. The immersion frequency was compared at intervals of 3, 6, and 12 h. In each case, the PLBs were immersed for 5 or 10 min. This experimental setup allowed for the evaluation of how different immersion intervals impacted the growth process of the PLBs. The TIB system was maintained at 25 ± 1 °C and a humidity of 51 ± 2% under white LED light at an intensity of 40 µmol m^−2^ s^−1^ for 16 h day^−1^ for 1 month.

### 3.3. Root Induction Process

Samples of *Grammatophyllum speciosum* Blume with 2.0 cm shoot lengths were cultured in six different rooting media. RT1: ½ MS medium supplemented with 0.5 mg L^−1^ of NAA and 1 mg L^−1^ of BAP, RT2: ½ MS medium supplemented with 0.5 mg L^−1^ of NAA; RT3: MS medium supplemented with 0.5 mg L^−1^ of NAA and 1 mg L^−1^ of BAP; RT4: MS medium supplemented with 0.5 mg L^−1^ of NAA; RT5: VW medium [37] supplemented with 150 mL L^−1^ of fresh coconut water, 100 g L^−1^ of fresh banana and 50 g L^−1^ of fresh potato; RT6: VW medium. The culture media contained 20 g L^−1^ of sucrose and 1 g L^−1^ of charcoal and was solidified using 7.2 g L^−1^ of agar. The pH of the media was adjusted to 5 using 1 M HCl and NaOH. The cultures were maintained at 25 ± 1 °C and humidity of 51 ± 2% under white LED light at an intensity of 40 µmol m^−2^ s^−1^ for 16 h day^−1^ for 1 month.

### 3.4. Morphological Evaluation

Different stages of development in *Grammatophyllum speciosum* Blume were observed every 7 days in the culture. Fresh explants were selected and observed under a stereo microscope with a camera (Olympus SZ 40) (Olympus Corporation, Center Valley, PA, USA) for morphological studies. The morphological parameters of the explant, including the number of shoots and roots as well as the maximum length of shoots, roots, and leaves, were recorded following cultivation in different culture media. The fresh growth index was assessed after culture using the following equation.(1)$$\mathrm{Fresh}\;\mathrm{Growth}\;\mathrm{Index}\;(\mathrm{FGI})=\frac{\left[\mathrm{Final}\;\mathrm{Fresh}\;\mathrm{Weight}\;-\;\mathrm{Initial}\;\mathrm{Fresh}\;\mathrm{Weight}\right]}{\mathrm{Initial}\;\mathrm{Fresh}\;\mathrm{Weight}}$$

### 3.5. Sodium Chloride Stress Treatments

One-year-old plantlets of *Grammatophyllum speciosum* Blume were cultured in ½ MS liquid medium supplemented with 20 g L^−1^ sucrose, 1 g L^−1^ activated charcoal, and sodium chloride (NaCl) at 0, 50, 100 or 200 µM to evaluate the total phenolic content in the in vitro culture for 72 h. The cultures were maintained at 25 ± 1 °C and humidity of 51 ± 2% under white LED light at an intensity of 40 µmol m^−2^ s^−1^ for 16 h day^−1^ using the TIB system (IF4, 10 min immersion time every 3 h).

### 3.6. Preparation of the Crude Extract

The NaCl-treated plantlets were collected and dried in a hot-air oven at 60 °C for 2 days. The dried samples (100 mg each) were separately ground to a fine powder and soaked in 100 mL of ethyl alcohol. The extracts were incubated at room temperature for 3 days and shaken occasionally. After incubation, the extracts were passed through Whatman^TM^ No.1 qualitative filter paper (Merck KGaA, Darmstadt, Germany) and concentrated at 40 °C using a speed vacuum concentrator (Eppendorf, Hamburg, Germany). The obtained semisolid extracts were stored at −20 °C.

### 3.7. Quantification of Total Phenolic Content

The total phenolic content was determined using the Folin–Ciocalteu colorimetric method [76]. The reaction mixture was created in a 96-well plate by combining 20 µL of each extract (dissolved in ethyl alcohol) and 100 µL of 10% (*v*/*v*) Folin–Ciocalteu reagent (Merck KGaA, Darmstadt, Germany). The mixture was incubated at room temperature for 3 min. Then, 80 µL of 1 M sodium carbonate (Na_2_CO_3_) was added and incubated at room temperature in the dark for 20 min. The absorbance was measured at 765 nm using a microplate reader (Tecan, Männedorf, Switzerland). Gallic acid (Sigma-Aldrich, St. Louis, MO, USA) at 0, 0.0125, 0.025, 0.05, 0.1, or 0.2 mg mL^−1^ was used as a calibrated standard. The results were expressed in milligrams of gallic acid equivalent per gram of dry weight (mg GAE g^−1^ DW).

### 3.8. DPPH Radical Scavenging Assay

The antioxidant activity of the extracts was determined using 2,2-diphenyl-2-picrylhydrazyl (DPPH), according to the method of Herald [77]. A sample (20 µL) of an extract was mixed with 200 µL of 1.5 mM DPPH solution in ethyl alcohol and incubated at room temperature in the dark for 30 min. The absorbance was measured at 517 nm using a microplate reader. Ascorbic acid (Fisher Scientific, Loughborough, UK) at 0, 2.5, 5, 10, 20, 40, 80, or 160 µg mL^−1^ was used as a calibrated standard. The results were expressed as the percentage of DPPH scavenging activity using the following equation.DPPH scavenging activity (%) = [(A0 − A1)/A0] × 100(2)
where A0 is the absorbance of the control and A1 is the absorbance of the extract.

### 3.9. Ferric-Reducing Antioxidant Power Assay

The ferric-reducing antioxidant power (FRAP) assay was carried out using the method described by Benzie and Strain [78], with minor modifications for use in a 96-well microplate. The FRAP reagent was prepared by mixing 300 mM sodium acetate buffer (pH 3.6), 10 mM 2,4,6-tripyridyl-s-triazine (TPTZ) in 40 mM hydrochloric acid, and 20 mM ferric chloride hexahydrate (FeCl_3_·6H_2_O) solution in the ratio 10:1:1 at 37 °C in the dark. Then, 20 µL of each extract was mixed with 280 µL of freshly prepared working FRAP reagent. The reaction mixture was incubated at 37 °C in the dark for 30 min. The absorbance of the ferrous-tripyridyltriazine (Fe (II)-TPTZ) complex blue color was measured at 593 nm using a microplate reader. Ascorbic acid (at 0, 62.5, 125, 250, 500, or 1000 µM was used to construct a calibration curve. The FRAP value was expressed in micromoles of ascorbic acid equivalent per gram of dry weight (µM AAE g^−1^ DW).

### 3.10. Antibacterial Activity Testing

*Staphylococcus aureus* (DMST 8840), *Staphylococcus epidermidis* (DMST 15505), and *Propionibacterium acnes* (DMST 14916) were obtained from the Herbal and Bioactive Substances Technology Laboratory, Kasetsart Agricultural and Agro-Industrial Product Improvement Institute (KAPI), Bangkok, Thailand. The two aerobic bacteria (*S. aureus* and *S. epidermidis*) were cultured on tryptone soy agar at 37 °C for 24 h, while the aero-tolerant anaerobic bacterium (*P. acnes*) was cultured on brain heart infusion agar at 37 °C for 72 h under anaerobic conditions. The colonies were evenly suspended in 0.85% NaCl and diluted to a turbidity equivalent to McFarland No. 0.5 (108 colony forming units mL^−1^).

The minimum inhibitory concentration (MIC) and minimum bactericidal concentration (MBC) were determined using the two-fold serial microdilution method [79]. The diluted extracts (in dimethyl sulfoxide) were prepared at concentrations in the range 0.05–102.4 mg mL^−1^ with sterile Mueller-Hinton broth (for *S. aureus* and *S. epidermidis*) and brain heart infusion broth (for *P. acnes*) before adding the bacterial suspensions (100 µL each) into the 96-well microtiter plates. *S. aureus* and *S. epidermidis* were incubated at 37 °C for 24 h, while *P. acnes* was incubated at 37 °C under anaerobic conditions for 72 h. The bacterial suspensions were used as positive controls and the crude extracts in broth were used as negative controls. The MIC values were defined as the lowest concentrations of the crude extracts that inhibited the growth of microorganisms. The MBC values were determined by inoculating a broth with no visible bacterial growth onto agar plates and were defined as the lowest concentration of the extracts needed to kill each bacterium.

### 3.11. Statistical Analysis

Comparisons of the data between the treatments were carried out based on Tukey’s multiple comparisons test, with significance tested at *p* ≤ 0.05 using GraphPad Prism 6 software (GraphPad Software, Inc., San Diego, CA, USA). Results were expressed as mean ± standard deviation (SD) values.

## 4. Conclusions

In the current research, we successfully established a highly effective method for micro-propagation of *Grammatophyllum speciosum* Blume using the Temporary Immersion Bioreactor (TIB) system. This research is the first to report the successful application of TIB technology for cultivating this plant. We identified an optimal growth medium that supports both shoot and root induction, leading to significantly enhanced production under controlled, sterile conditions. Furthermore, our findings demonstrate that NaCl stress stimulates *G. speciosum* to produce a greater amount of phenolic compounds, directly enhancing its antioxidant capacity. The extracted compounds demonstrated promising antibacterial activity against the bacteria that cause acne. This is a significant finding, given that *G. speciosum* extract is already used in traditional Thai treatments.

This TIB technology provides a strategic advantage for the sustainable and efficient production of high-quality secondary metabolites. The next phase of our research will focus on a deeper evaluation of the biological activities of the *G. speciosum* extracts, specifically their antibacterial properties. Our ultimate vision is to see these findings lead to the development of new, medically viable products, promoting the use of Thai medicinal herbs in the global pharmaceutical industry. This TIB process offers a promising avenue for largescale production, unlocking the full potential of *G. speciosum* for pharmaceutical and cosmetic applications.

## Figures and Tables

**Figure 1 plants-14-03083-f001:**
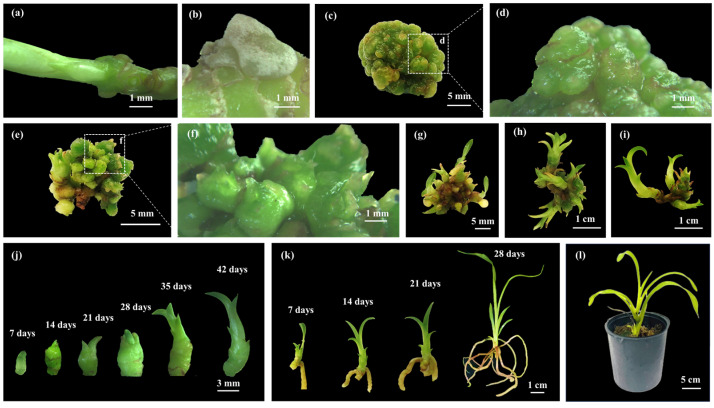
Morphological development of *Grammatophyllum speciosum.* Blume (**a**) Nodal segment of explant. (**b**) Formation of embryogenic calli from nodal segment after culture in T2 medium. (**c**,**d**) Top view of explant showing formation of PLBs from shoot apex after 14 days of culture in T2 medium. (**e**,**f**) PLB formations were clearly detected and showed well-grown explant surface after 21 days of culture in T2 medium. (**g**–**i**) Shoot development in different growth stages on explant after 28 days (**g**), 35 days (**h**), and 42 days (**i**) of culture in T2 medium. (**j**) Shoots in different growth stages from 7 days to 42 days after subculture. (**k**) Plantlet in different growth stages showing root development in root induction medium (RT5) from 7 days to 28 days of culture. (**l**) Plantlets of *G. speciosum* after transplantation for 3 months. Representative images from large-scale propagation experiments are shown.

**Figure 2 plants-14-03083-f002:**
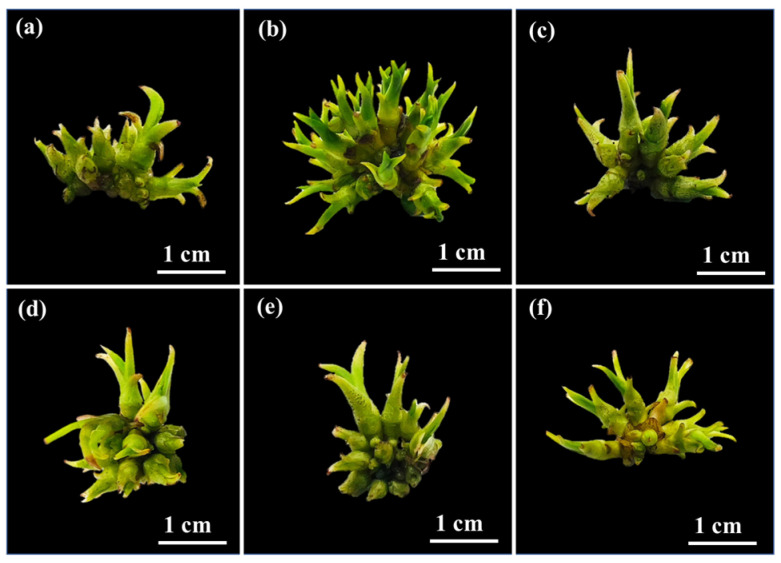
Effect of plant growth regulators on shoot proliferation of *Grammatophyllum speciosum* Blume after 42 days of culture. (**a**) T1 medium, ½ MS. (**b**) T2 medium, ½ MS supplemented with 1 mg L^−1^ of NAA and 0.5 mg L^−1^ of BAP. (**c**) T3 medium, ½ MS supplemented with 1 mg L^−1^ of NAA and 1 mg L^−1^ of BAP. (**d**) T4 medium, ½ MS supplemented with 2 mg L^−1^ of NAA and 0.5 mg L^−1^ of BAP. (**e**) T5 medium, ½ MS supplemented with 2 mg L^−1^ of NAA and 1 mg L^−1^ of BAP. (**f**) T6 medium, ½ MS supplemented with 2 mg L^−1^ of NAA and 2 mg L^−1^ of BAP. Representative images were obtained from ten biological replicates.

**Figure 3 plants-14-03083-f003:**
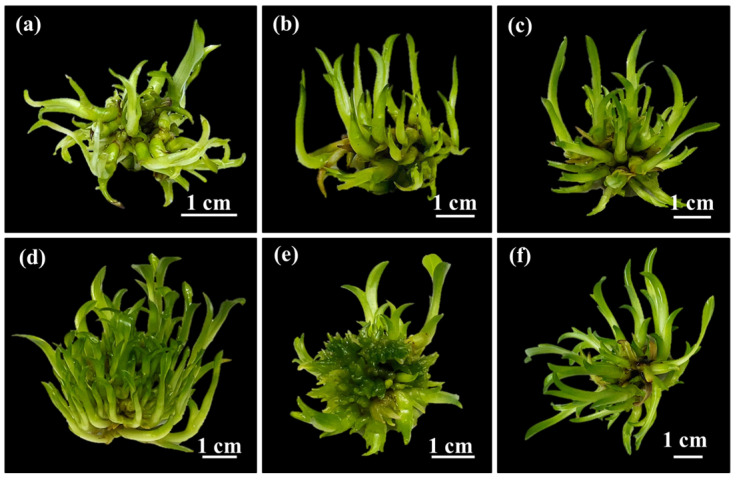
Shoot multiplication from protocorm-like bodies under different immersion times and frequencies within TIB system after 1 month. (**a**) IF1: 5 min immersion every 3 h. (**b**) IF2: 5 min immersion every 6 h. (**c**) IF3: 5 min immersion every 12 h. (**d**) IF4: 10 min immersion every 3 h. (**e**) IF5: 10 min immersion every 6 h. (**f**) IF6: 10 min immersion every 12 h. Representative images were obtained from six biological replicates.

**Figure 4 plants-14-03083-f004:**
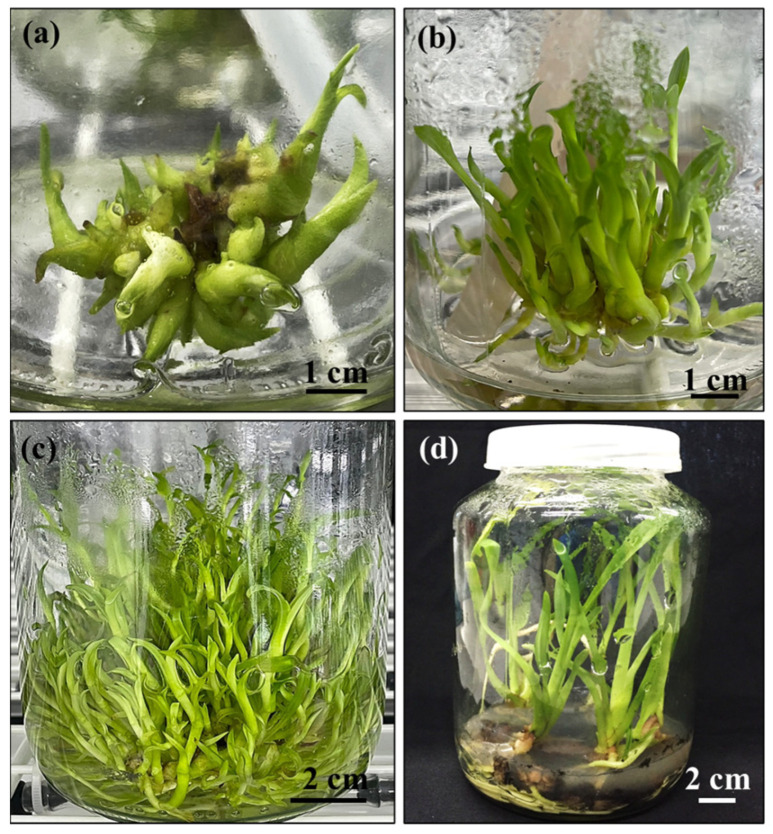
Overview of *Grammatophyllum speciosum* Blume propagation. (**a**) Development of PLBs for shoot induction in TIB system (IF4) on T2 medium after 7 days. (**b**) Shooting in TIB system (IF4) on T2 medium after 1 month. (**c**) Multiplication of shoots within TIB system (IF4) on T2 medium after 3 months. (**d**) Plantlets on RT5 medium after 3 months. Representative images from large-scale propagation experiments are shown.

**Figure 5 plants-14-03083-f005:**
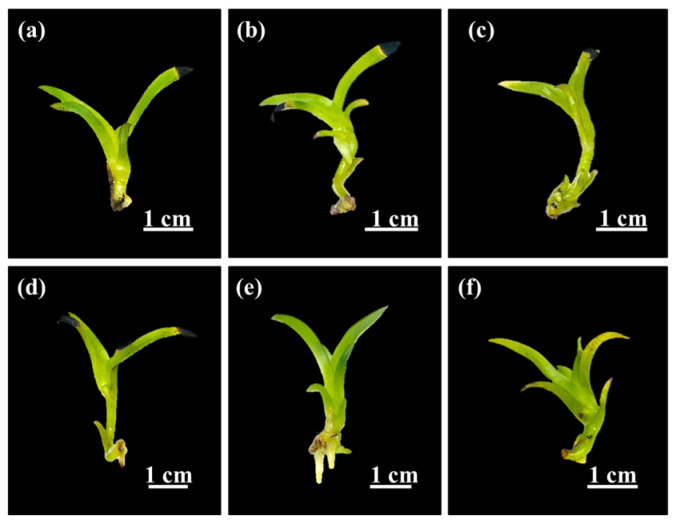
The plantlets of *Grammatophyllum speciosum* Blume following a 1-month cultivation period in six different rooting mediums ((**a**–**f**): RT1–RT6). Representative images were obtained from ten biological replicates.

**Figure 6 plants-14-03083-f006:**
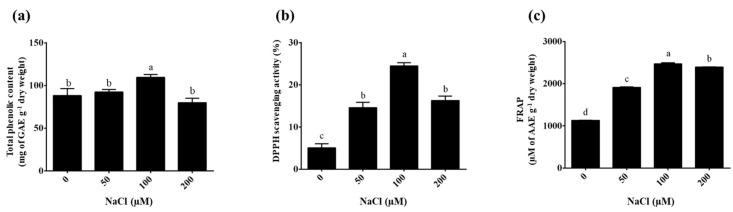
Total phenolic content and antioxidant activity levels of *Grammatophyllum speciosum* Blume extracts under salinity treatments for 3 days. (**a**) Total phenolic content. (**b**) Percentage of DPPH scavenging activity. (**c**) FRAP values. Data represent mean ± SD values of 10 replications. Different lowercase letters indicate significant (*p* < 0.05) differences among NaCl concentrations based on Tukey’s multiple comparisons test.

**Figure 7 plants-14-03083-f007:**
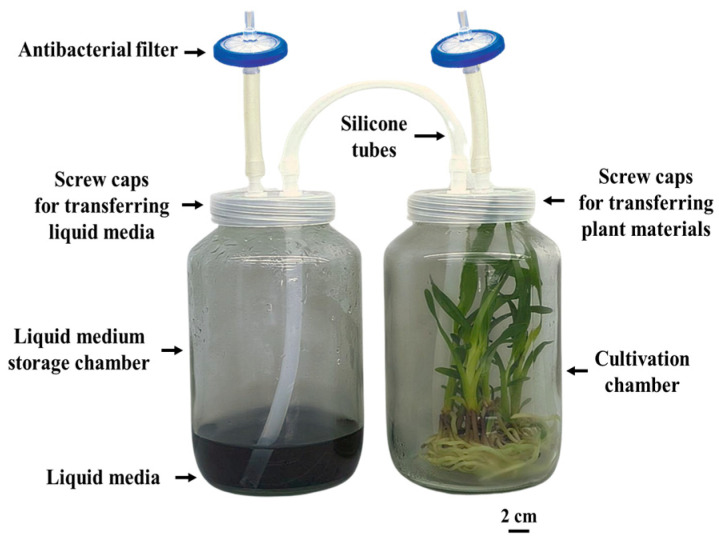
Shoot proliferation in glass chamber of temporary immersion bioreactor with twin flask system.

**Figure 8 plants-14-03083-f008:**
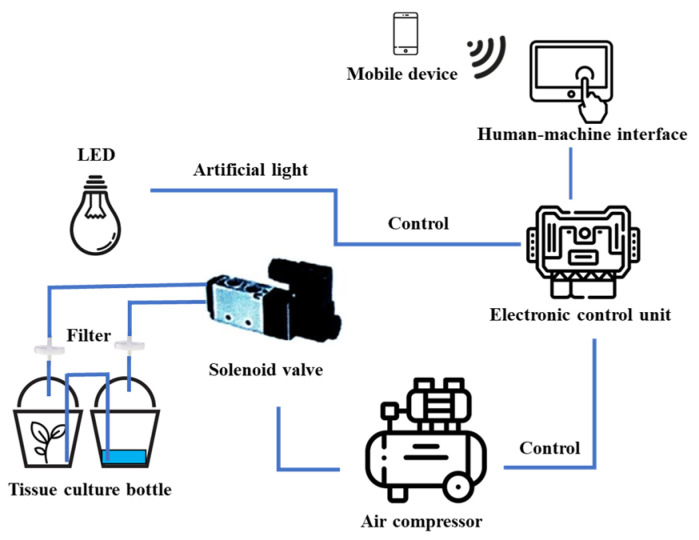
Schematic diagram of temporary immersion bioreactor system integrated with Internet of Things (IoT) technology.

**Table 1 plants-14-03083-t001:** Effect of PLB induction media on morphological development of *Grammatophyllum speciosum* Blume. Values are the means of ten replicates ± SD. Different letters indicate significant difference between treatments as determined by two-way ANOVA followed by Tukey’s multiple comparisons test. Different lowercase letters indicate significant (*p* ≤ 0.05) differences among treatment. “ns” indicates not statistically significant.

Treatment	Hormone (mg L^−1^)		No. Shoots/Explant			Fresh Growth Index		
	BAP	NAA	28 Days	35 Days	42 Days	28 Days	35 Days	42 Days
T1	-	-	9.00 ± 1.73 ^b^	18.33 ± 1.53 ^b^	19.33 ± 1.15 ^b^	0.63 ± 0.11 ^ns^	5.44 ± 0.46 ^b^	6.69 ± 0.40 ^b^
T2	1.0	0.5	16.00 ± 1.73 ^a^	32.33 ± 2.52 ^a^	35.00 ± 3.00 ^a^	1.00 ± 0.34 ^ns^	8.78 ± 2.09 ^a^	9.92 ± 1.31 ^a^
T3	1.0	1.0	7.67 ± 0.58 ^b c^	18.00 ± 2.00 ^b^	18.67 ± 2.08 ^b^	0.55 ± 0.15 ^ns^	5.14 ± 0.77 ^b^	6.41 ± 0.77 ^bc^
T4	2.0	0.5	4.33 ± 2.31 ^c^	10.67 ± 1.15 ^c^	12.33 ± 1.53 ^c^	0.23 ± 0.07 ^ns^	4.44 ± 1.07 ^bc^	4.88 ± 0.60 ^cd^
T5	2.0	1.0	8.00 ± 1.00 ^bc^	12.67 ± 2.08 ^c^	13.67 ± 1.53 ^c^	0.66 ± 0.12 ^ns^	4.89 ± 0.19 ^bc^	5.83 ± 0.06 ^bc^
T6	2.0	2.0	5.67 ± 1.15 ^bc^	6.33 ± 1.15 ^d^	6.67 ±0.58 ^d^	0.38 ± 0.06 ^ns^	3.17 ± 1.11 ^c^	3.82 ± 0.77 ^d^

**Table 2 plants-14-03083-t002:** Effect of different immersion times and frequencies on shoot multiplication of *Grammatophyllum speciosum* Blume after 1 month in TIB system. Values are means of six replicates ± SD. Different letters indicate significant differences between treatments. Each response variable was analyzed using one-way ANOVA, and significant differences among treatment groups were determined by Tukey’s multiple comparisons test (*p* ≤ 0.05).

Treatment	Immersion Frequency		No. Shoots/Explant	Shoot Height (cm)	Fresh Growth Index
	Frequency/Day	Immersion			
IF1	8	5 min, every 3 h	20.00 ± 0.82 ^d^	2.77 ± 0.31 ^bc^	0.23 ± 0.02 ^c^
IF2	4	5 min, every 6 h	38.00 ± 2.94 ^c^	1.93 ± 0.12 ^bc^	2.12 ± 0.63 ^b^
IF3	2	5 min, every 12 h	33.00 ± 3.27 ^cd^	1.10 ± 0.33 ^cd^	2.32 ± 0.50 ^b^
IF4	8	10 min, every 3 h	127.00 ± 2.16 ^a^	5.00 ± 0.51 ^a^	4.26 ± 0.52 ^a^
IF5	4	10 min, every 6 h	95.33 ± 7.59 ^b^	1.27 ± 0.12 ^d^	3.21 ± 0.59 ^ab^
IF6	2	10 min, every 12 h	47.33 ± 5.56 ^c^	3.70 ± 0.45 ^b^	3.05 ± 0.50 ^ab^

**Table 3 plants-14-03083-t003:** Effect of root induction media on morphological characteristic of *Grammatophyllum speciosum* Blume after 1 month. Values are means of ten replicates ± SD. Different letters indicate significant difference between treatments based on one-way ANOVA followed by Tukey’s multiple comparisons test (*p* ≤ 0.05). nd means ‘not detected’. ns means ‘not statistically significant’. * means VW medium supplemented with 150 mL L^−1^ of fresh coconut water, 100 g L^−1^ of fresh banana and 50 g L^−1^ of fresh potato.

Treatment	Medium	Hormone (mg L^−1^)		No. Roots /Explant	Length of Roots (cm)	Fresh Growth Index	Shoot Height (cm)	No. Leaves /Explant	Length of Leaves (cm)
	½ MS	BAP	NAA						
RT1	½ MS ½ MS	1.0	0.5	1.00 ± 0.00	0.40 ± 0.10 ^b^	1.17 ± 0.50 ^c^	2.35 ± 0.15 ^ns^	4.50 ± 0.50 ^ns^	1.25 ± 0.05 ^ns^
RT2		-	0.5	0.50 ± 0.50 ^b^	0.20 ± 0.20 ^b^	2.42 ± 0.08	3.45 ± 0.35 ^ns^	3.50 ± 0.50 ^ns^	2.20 ± 0.30 ^ns^
RT3	MS	1.0	0.5	nd	nd	1.75 ± 0.08 ^b^	2.65 ± 0.15 ^ns^	4.50 ± 0.50 ^ns^	0.95 ± 0.15 ^ns^
RT4	MS	-	0.5	nd	nd	1.83 ± 0.17 ^b^	2.75 ± 0.25 ^ns^	3.50 ± 0.50 ^ns^	1.75 ± 0.25 ^ns^
RT5	VW *	-	-	2.50 ± 0.50 ^a^	1.35 ± 0.15 ^a^	3.92 ± 0.25 ^a^	2.85 ± 0.05 ^ns^	4.50 ± 0.50 ^ns^	2.25 ± 0.15 ^ns^
RT6	VW *	-	-	0.50 ± 0.50 ^b^	0.50 ± 0.50 ^b^	3.00 ± 0.33 ^ab^	3.35 ± 0.15 ^ns^	3.50 ± 0.50 ^ns^	0.95 ± 0.25 ^ns^

**Table 4 plants-14-03083-t004:** Antibacterial activity of *Grammatophyllum speciosum* Blume extracts.

Bacterium	MIC (mg mL^−1^)	MBC (mg mL^−1^)
*Staphylococcus aureus*	12.8	25.6
*Staphylococcus epidermidis*	25.6	51.2
*Propionibacterium acnes*	6.4	12.8

## Data Availability

Data are contained within the article.

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
