# Peer review of "Production of Bioactive Compounds in Grammatophyllum speciosum Blume Using Bioreactor Cultures Under Elicitation with Sodium Chloride"

_plants, 2025, doi:10.3390/plants14193083_

Round 1
Reviewer 1 Report
Comments and Suggestions for Authors
Dear Authors,
The manuscript entitled: "Bioactive Compound Production of Grammatophyllum speciosum Blume Using Bioreactor Cultures Under Elicitation with Sodium Chloride" presented for the assessment, in my opinion is generally written in the correct form. The information contained in it could be cognitive and application significant and I think the presented data and conclusions could interest many researchers and readers after some refilling. Although the work is very interesting and of application value I think that You should take into account some modification of this article. I recommend publishing it in "Plants” after a minor revision.
Several specific comments:
Title: “Bioactive Compound Production of Grammatophyllum speciosum Blume Using Bioreactor Cultures Under Elicitation with Sodium Chloride” – I suggest to add “s” in the word “compounds”
Abstract:
Line 14 – 15: “This research propagated G. speciosum from vegetative organs grown under different NaCl-stressed conditions.” - from such a formulated statement it may follow that NaCl was a factor in the propagation of G. speciosum. If this is not the authors' intention (the next sentence proves this), it should be verified.
Line 17: “…proved to be a suitable medium for…” – I suggest to exchange into “… proved to be more suitable or more effective medium for…..”.
Line 18 - 19: “The protocorm-like bodies were transferred into a temporary immersion bioreactor (TIB) system.” - Does this mean that the protocorm-like bodies were primary explants in the experiment? Please clarify.
Line 23: “…Sodium chloride…” – sodium chloride
Introduction:
Line 39: “waan phet cha hueng” – if this is possible it is good to explain the meaning of this name.
Line 45-46: “…These therapeutic effects result from the biological compounds produced throught the plant’s metabolism.” - that is obvious, please give some example proved by citations
Line 49: “…produces disease-free plants..” – I do not think it is true, more proper is to say “….pathogen-free plants”, should be verified.
Line 50: “In vitro” – In vitro - italic
Line 59: “……and other economic plants..” - it would be good to give some examples of economically important species and their uses
In the Introduction section, it would be good to explain what protocorm-like bodies are, what their importance is for orchids, including the species under study, whether mycorrhiza occurs within this species, etc.
The Introduction does not present the purpose of the investigation clearly as well as there is not hipotesis included. This omission should be corrected and placed at the end of this chapter.
Results and Discussion:
Line 144 – 163: Effect of different immersion times and frequencies on shoot multiplication
This part requires a deeper discussion with an explanation of the hypothetical or real reason for the state of affairs described by the Authors, please complete it.
Line 183 – 183: “The composition of the MS medium consisted of both macro- and micro-nutrients, as outlined by Murashige and Skoog[36]”. – This is not necessary; a simple citation is sufficient. Such a statement adds nothing new and should be included in the M&M.
Line 230: “…salinity acts as an elicitor to induce the accumulation of phenolic compounds in plants.” – This is too strong and categorical a statement. I think it should be changed to state that, within a specific range, and within a physiological range, salinity has such an effect.,
Line 255: “…The results indicated that salt stress could ……” – again, like before: This is too strong and categorical a statement. I think it should be changed to state that, within a specific range, and within a physiological range, salinity has such an effect.,
Lines 269, 270, 284, 285: P. acnes, S. aureus and S. epidermidis, P. acnes - The full names of the bacteria should be provided.
Materials and Methods:
Line 292: Grammatophyllum speciosum – G. speciosum
Conclusions:
These part should be reworded and include the main results/achievements of the planned project, culminating in a project vision for the future.
Author Response
Dear Reviewer,
Thank you very much for taking the time to review the article Bioactive Compound Production of Grammatophyllum speciosum Blume Using Bioreactor Under Elicitation with Sodium Chloride. Please find the detailed responses in the resubmitted files. Our response and revisions are in the response letter as below. Please see the attachment.
Thank you again for your time and consideration
Sincerely,
Authors

Reviewer 2 Report
Comments and Suggestions for Authors
The manuscript (MS) presents research results regarding the establishment of a suitable protocol for micropropagation of G. speciosum focusing on the effect of growth regulators (NAA and BAP) for obtaining and optimizing shoot culture from Protocorm-like bodies (PLBs), the effect of immersion time for shoot multiplication in a temporary immersion bioreactor (TIB) and the effect of culture medium on root induction. The study was also focused on defining culture conditions for inducing enhancement of total phenolic content (TPC) and other antioxidant response scavenging activity DPPH and FRAP and applying a NaCl and the extract showing the highest total phenolic content was used for studying antibacterial activity against acne pathogens.
The main critical points of this MS are that the innovative aspects presented are limited. Two relevant previous studies have already reported suitable and efficient protocols for this species concerning either in vitro culure on solid and liquid medium (Sopalum et al. 2010 – Plant Cell Tissue Organ Cult.) a in the liquid culture in TIB, the total phenolic content (TPC), other antioxidant response scavenging activity DPPH and FRAP and the antibacterial activity on several bacteria including S. aureus and S. epidermidis (Thamrongwatwongsa et al. 2024 – Heliyon). Taking into consideration those previous studies, I think, it is necessary to highlight wich are the really innovative aspects and to perform further studies.
On my opinion, the innovative aspects relay mainly on the results of the part of study on the response in terms of increasing of TPC and scavenging activity of the salt treated in vitro growing G. speciosum and the study of the extract of NaCl treated shoots on against acne pathogens. However, considering the previous study of Thamrongwatwongsa et al. (2024) above reported, I also suggest performing some more experimental work to compare that the extracts from salt treated explants are more effective in antibacterial activity than the not salt treated explants.
Also, an other innovative aspect could be that of the effect of VW medium with or without coconut water, banana and potato. On the other hand, concerning the comparison of MS and ½ MS with VW medium some more work, on my opinion, has to be performed using the same growth regulators combinations with all salt combinations.
Finally, I am afraid to have to conclude that the manuscript is not suitable for publication on this journal.
Other points:
The title, on my opinion, should be improved highlighting better the content of the studies performed.
Check the description the results according to the letters based on Tukey’s multiple comparisons test indicating different significant means reported in Table 1: Lines 104 – 105: The sample with the greatest number of shoots had the highest fresh growth index. 104 The T4 medium had the lowest shoot number; Lines 107 and 108: while the T3, T5, and T6 media formed similar shoot numbers to T1 but had 107 different fresh growth index values.
I suggest adding more details in most of the legends of tables and figures. They are supposed to be self-explaining and include the explication of the abbreviation included in the tables and figures.
Author Response
Dear Reviewer,
Thank you very much for taking the time to review the article Bioactive Compound Production of Grammatophyllum speciosum Blume Using Bioreactor Under Elicitation with Sodium Chloride. Please find the detailed responses below.
-Point-by-point response to comments and suggestions
Comments 1: The manuscript (MS) presents research results regarding the establishment of a suitable protocol for micropropagation of G. speciosum focusing on the effect of growth regulators (NAA and BAP) for obtaining and optimizing shoot culture from Protocorm-like bodies (PLBs), the effect of immersion time for shoot multiplication in a temporary immersion bioreactor (TIB) and the effect of culture medium on root induction. The study was also focused on defining culture conditions for inducing enhancement of total phenolic content (TPC) and other antioxidant response scavenging activity DPPH and FRAP and applying a NaCl and the extract showing the highest total phenolic content was used for studying antibacterial activity against acne pathogens.
The main critical points of this MS are that the innovative aspects presented are limited. Two relevant previous studies have already reported suitable and efficient protocols for this species concerning either in vitro culure on solid and liquid medium (Sopalum et al. 2010 – Plant Cell Tissue Organ Cult.) a in the liquid culture in TIB, the total phenolic content (TPC), other antioxidant response scavenging activity DPPH and FRAP and the antibacterial activity on several bacteria including S. aureus and S. epidermidis (Thamrongwatwongsa et al. 2024 – Heliyon). Taking into consideration those previous studies, I think, it is necessary to highlight wich are the really innovative aspects and to perform further studies.
On my opinion, the innovative aspects relay mainly on the results of the part of study on the response in terms of increasing of TPC and scavenging activity of the salt treated in vitro growing G. speciosum and the study of the extract of NaCl treated shoots on against acne pathogens. However, considering the previous study of Thamrongwatwongsa et al. (2024) above reported, I also suggest performing some more experimental work to compare that the extracts from salt treated explants are more effective in antibacterial activity than the not salt treated explants.
Also, another innovative aspect could be that of the effect of VW medium with or without coconut water, banana and potato. On the other hand, concerning the comparison of MS and ½ MS with VW medium some more work, on my opinion, has to be performed using the same growth regulators combinations with all salt combinations.
Finally, I am afraid to have to conclude that the manuscript is not suitable for publication on this journal.
Response: We thank the reviewer for the thoughtful comments. We would like to clarify and emphasize the novel aspects and scientific contribution of our study, in relation to previous work:
- Sopalum et al. (2010) induced PLBs from shoot tips of speciosum using ½ MS liquid medium and investigated the effects of various carbon sources and chitosan on PLB growth and regeneration. Their study primarily focused on optimizing PLB induction and relative growth in liquid and solid media. In contrast, the present study adopted a temporary immersion bioreactor (TIB) system to propagate PLBs, aiming to enhance clonal multiplication efficiency. While the induction step followed the use of sucrose as a carbon source similar to Sopalum et al., the integration of TIB represents an innovative approach for scaling up propagation, highlighting a clear distinction in the experimental objectives between the two studies. Moreover, Plantlets obtained via the TIB system not only allowed rapid clonal propagation but also proved effective in accumulating bioactive compounds under NaCl-induced stress, demonstrating the dual benefit of this propagation strategy.
- Compared to previous work by Thamrongwatwongsa et al. (2024) several key differences and innovations are highlighted. First, NaCl was applied as an elicitor in our study to stimulate the accumulation of phenolic and antioxidant compounds, whereas Thamrongwatwongsa et al. did not include salt stress in their protocol. Second, the temporary immersion bioreactor (TIB) system was employed in our work specifically for the efficient clonal propagation of protocorm-like bodies (PLBs) to produce healthy and robust plantlets, while Thamrongwatwongsa et al. used TIB only for routine tissue culture without stress treatment. Importantly, the primary aim of our work was to establish an efficient TIB-based micropropagation system, producing healthy plantlets that can serve as reliable raw material for bioactive compound production, represents a novel contribution beyond previous studies.
Other points:
The title, on my opinion, should be improved highlighting better the content of the studies performed.
Check the description the results according to the letters based on Tukey’s multiple comparisons test indicating different significant means reported in Table 1: Lines 104 – 105: The sample with the greatest number of shoots had the highest fresh growth index. 104 The T4 medium had the lowest shoot number; Lines 107 and 108: while the T3, T5, and T6 media formed similar shoot numbers to T1 but had 107 different fresh growth index values.
I suggest adding more details in most of the legends of tables and figures. They are supposed to be self-explaining and include the explication of the abbreviation included in the tables and figures.
Comments 1: …Taking into consideration those previous studies, I think, it is necessary to highlight which are the really innovative aspects and to perform further studies.
Response 1: We appreciate the reviewer’s valuable comment regarding the novelty of our work. In the revised manuscript, we have highlighted more clearly that the innovative aspect of this study is the use of a temporary immersion bioreactor (TIB) system for PLB production. This approach not only improves the efficiency of plantlet multiplication but also provides high-quality plant materials with strong potential for bioactive compound production. This information has been added to the last paragraph of the Introduction section (line 90-95).
Comments 2: …I also suggest performing some more experimental work to compare that the extracts from salt treated explants are more effective in antibacterial activity than the not salt treated explants.
Response 2: We thank the reviewer for the insightful suggestion. Although additional experiments comparing antibacterial activities of salt-treated vs. non-treated explants were not performed in this work, the increased phenolic and antioxidant contents in salt-treated explants already support the potential of enhanced antibacterial properties. Since the primary aim of this study was to optimize PLB proliferation in TIB and to provide preliminary evidence of bioactivity, a comprehensive antibacterial comparison will be explored in future research.
Comments 3: …another innovative aspect could be that of the effect of VW medium with or without coconut water, banana and potato. On the other hand, concerning the comparison of MS and ½ MS with VW medium some more work, on my opinion, has to be performed using the same growth regulators combinations with all salt combinations.
Response 3: We sincerely appreciate your valuable suggestion regarding the comparison between MS, ½ MS, and VW medium under identical growth regulator and salt combinations. While we fully agree that such experiments would provide additional insights, they fall beyond the scope and objectives of the present study, which primarily focused on optimizing TIB culture for PLB proliferation and evaluating the effect of NaCl stress on phenolic accumulation and antioxidant activity. Nevertheless, this point has been acknowledged in the revised manuscript (page 7, lines 231–235).
Comments 4: Other points: The title, on my opinion, should be improved highlighting better the content of the studies performed.
Response 4: This change can be found – page number 1
Response 5: This change can be found – page number 3-4; line 121-123, 127-131
Comments 6: …adding more details in most of the legends of tables and figures. They are supposed to be self-explaining and include the explication of the abbreviation included in the tables and figures.
Response 6: We thank the reviewer for the suggestion. This change can be found – line 110, 139, 145-148, 199-202, 203-207, 242, 247-251
-Response to Comments on the Quality of English Language
The article Bioactive Compound Production of Grammatophyllum speciosum Blume Using Bioreactor Cultures Under Elicitation with Sodium Chloride has undergone English language editing by MDPI. The text has been checked for correct use of grammar and common terms, and edited to a level suitable for reporting research in a scholarly journal.

Reviewer 3 Report
Comments and Suggestions for Authors
The manuscript looks at phenolics production in regenerated plants of G. speciosum. The authors examine a variety of protocols for enhancing shoot and root production and subsequent plant regeneration. Separately, the phenolics production rate was examined in whole plantlets as a function of salt (elicitor) concentration. The manuscript can benefit from a number of clarifications around the methods and data analysis.
- The formation of PLBs is shown in Table 1 as a function of culture conditions and differences among cultures are assessed using Tukey's multiple comparisons test. This requires a preliminary ANOVA that is not specified by the authors. Are the data analyzed using a one-way (Treatments T1-T6) or a 2-way (BAP/NAA) model? Is the ANOVA test showing at least one of the means is different? If the test is a one-way ANOVA, the statements related to auxin and cytokinin ratios are not rigorously supported and the authors should carefully rephrase the results.
- The authors state (lines 126-127) "... the optimal proportion of BAP and NAA considerably promoted shoot generation in G. speciosum." This is not an "optimal" outcome as no model is specified for the responses of number of shoots nor fresh growth index as a function of NAA and BAP. It is simply the 'maximum' of these responses for the factor levels chosen.
- In Table 1, the column for Fresh Growth index at 28 days lacks alphabetic superscripts indicating significant differences.
- On lines 145-146, the authors state "both the immersion time and frequency were identified as crucial parameters affecting the shoot multiplication efficiency." This would require a two-way ANOVA but the authors do not specify that this is the case. The use of different letters in the columns of Table 2 suggest that the authors have used a one-way ANOVA on "Treatment" and then casually asserted "significant differences in growth parameters between immersion times of 5 min and 10 min." (lines 147-148). For the 2 immersion times, the statistical analysis should simply show if they are significant as factor levels in predicting responses. The authors also need to declare which responses they are referencing in statements that imply significance of factors. The authors need to carefully specify the statistical model they are using and discuss which factors are independent in the experimental design.
- Table 3 shows results for root induction and shoot morphological properties. There 6 responses, but only three seem to have a statistical analysis or perhaps the remaining three were not shown to be significant from the ANOVA? Here, again, the question is whether the analysis is simply based on Treatments (RT1-RT6) or if some use of medium/hormone designation is also a factor. This is important in that comments reflecting on the significance of the medium designation (MS vs VW) should be rigorously analyzed with the media identified as a factor. Otherwise, the comments on which media best supports the response its weak.
- The VW media is specified to contain potato and banana (lines 325- 326), but this needs to be better described. Does it really have raw potato or banana added to the media and if so what additional information on the types/cultivars can be provided?
- Section 3.2 should not use the word "Optimization" in the heading. There is no optimization going on here.
- Figure 8 can be removed. It is a trivial representation of the equipment that provides no additional insight beyond what the authors describe in the text.
- The sentence "The current effectively identified an appropriate medium ..." (line 415) is missing the sentence object - perhaps "The current research effectively identified an appropriate medium ..."
- Figure 5 has panels (a)-(f), but descriptions are only given for panels (a)-(d).
- The authors ultimately assessed phenolics production in regenerated plants as a function of NaCl. The protocol in Section 3.5 states "The 1-year-old plantlets of G. speciosum were cultured in ½ MS liquid medium supplemented with 20 g L-1 sucrose, 1 g L-1 activated charcoal, and sodium chloride (NaCl) at 0, 50, 100 or 200 µM to induce the total phenolic content from in vitro culture for 72 h. The cultures were maintained at 25 ± 1 °C and humidity of 51 ± 2% under white LED light at an intensity of 40 µmol m-2 s-1 for 16 h day-1 using the TIB system (IF4, 10 min immersion time every 3 h)." Were the 1 year-old plants propagated from the RT5? Was the TIB used for the whole year in propagating these stock plants prior to switching the media to 1/2 MS? Why switch to 1/2 MS from RT5? There seems to be a disconnect between the efforts that went into establishing a useful propagation protocol and the selection of plants used to look at phenolics production. The authors should clarify this.
Author Response

(The authors gave the same response as above.)

Reviewer 4 Report
Comments and Suggestions for Authors
In this study, the authors first established an in vitro regeneration system for Grammatophyllum speciosum, identifying the optimal culture medium formulation for cluster bud induction, a temporary immersion bioreactor (TIB) system for shoot multiplication, and a rooting medium for plantlets. Furthermore, the authors employed NaCl stress to induce secondary metabolite accumulation in plantlets of G. speciosum, thereby enhancing their antioxidant and antimicrobial activities. The findings offer valuable insights for the production of bioactive compounds derived from G. speciosum. However, several aspects of the manuscript remain unclear, as detailed below:
1. Why did the authors choose NaCl as an elicitor for enhancing bioactive compound production instead of commonly used phytohormones such as MeJA or SA? Moreover, the introduction does not provide a rationale for the use of NaCl.
2. While the authors reported that 100 µM NaCl stress resulted in the highest total phenolic content (TPC) and the strongest antioxidant and antimicrobial activities, data on the impact of NaCl stress on plantlet growth were not presented. If the stress significantly reduces biomass, the practicality of this elicitation strategy may be questionable. The authors should address this limitation either by providing additional data or discussing it in the context of the study’s shortcomings.
3. The statement in the abstract, “This research propagated G. speciosum from vegetative organs grown under different NaCl-stressed conditions,” is inconsistent with the actual methodology. NaCl stress was applied only at the plantlet stage, not throughout the propagation process.
Author Response
Dear Reviewer,
Thank you very much for taking the time to review the article Bioactive Compound Production of Grammatophyllum speciosum Blume Using Bioreactor Under Elicitation with Sodium Chloride. Please find the detailed responses below.
Point-by-point response to Comments and Suggestions
Comments 1: In this study, the authors first established an in vitro regeneration system for Grammatophyllum speciosum, identifying the optimal culture medium formulation for cluster bud induction, a temporary immersion bioreactor (TIB) system for shoot multiplication, and a rooting medium for plantlets. Furthermore, the authors employed NaCl stress to induce secondary metabolite accumulation in plantlets of G. speciosum, thereby enhancing their antioxidant and antimicrobial activities. The findings offer valuable insights for the production of bioactive compounds derived from G. speciosum. However, several aspects of the manuscript remain unclear, as detailed below:
- Why did the authors choose NaCl as an elicitor for enhancing bioactive compound production instead of commonly used phytohormones such as MeJA or SA? Moreover, the introduction does not provide a rationale for the use of NaCl.
Response 1: We appreciate the reviewer’s comment. NaCl was chosen as the elicitor because salinity stress is one of the most common abiotic stresses in plants and has been widely reported to enhance the accumulation of phenolics and other secondary metabolites. For example, Nag & Kumaria [47] demonstrated that NaCl can stimulate phenolic production in Vanda orchids. In general, salinity stress promotes the synthesis of antioxidant compounds as part of the plant’s defense against oxidative damage [lines 267–276]. MeJA or SA, which act as plant growth regulators and influence multiple aspects of plant development, NaCl was used in this study to impose a controlled stress condition. This provides a simple and environmentally relevant elicitation strategy to enhance the accumulation of bioactive compounds without confounding effects on growth. Furthermore, NaCl is inexpensive and easy to apply, making it a practical elicitor for large-scale culture systems.
Comments 2: While the authors reported that 100 µM NaCl stress resulted in the highest total phenolic content (TPC) and the strongest antioxidant and antimicrobial activities, data on the impact of NaCl stress on plantlet growth were not presented. If the stress significantly reduces biomass, the practicality of this elicitation strategy may be questionable. The authors should address this limitation either by providing additional data or discussing it in the context of the study’s shortcomings.
Response 2: We thank the reviewer for raising this important point. We acknowledge that biomass reduction under NaCl stress was not measured, which is a limitation of this study. Nevertheless, plantlets maintained healthy morphology throughout the NaCl treatment. Previous studies have shown that moderate salinity (100 µM NaCl) can stimulate secondary metabolite accumulation in orchids such as Vanda [Nag & Kumaria, 42]. Similarly, Dendrobium orchids respond to salinity through osmotic adjustment and sequestration of Na⁺ and Cl⁻ in the roots, preventing excessive ion accumulation in the shoots. These reports suggest that the applied NaCl concentration in our study was within a physiologically tolerable range, allowing enhanced phenolic production without severely affecting growth. We have now acknowledged this limitation in the Discussion (lines 272–275) and suggested that future studies should include quantitative biomass assessment to better evaluate the trade-off between growth and metabolite accumulation.
Comments 3: The statement in the abstract, “This research propagated G. speciosum from vegetative organs grown under different NaCl-stressed conditions,” is inconsistent with the actual methodology. NaCl stress was applied only at the plantlet stage, not throughout the propagation process.
Response 3: We thank the reviewer for the comment. The Abstract has been revised to clarify that NaCl stress was applied only at the plantlet stage, after propagation from vegetative organs (Abstract, lines 14-16).
Response to Comments on the Quality of English Language
The article Bioactive Compound Production of Grammatophyllum speciosum Blume Using Bioreactor Cultures Under Elicitation with Sodium Chloride has undergone English language editing by MDPI. The text has been checked for correct use of grammar and common terms, and edited to a level suitable for reporting research in a scholarly journal.

Round 2
Reviewer 3 Report
Comments and Suggestions for Authors
the authors have responded to the comments from the previous review and the manuscript has appropriate levels of detail to inform the readers. The authors have done a good job of editing and clarifying the experimental design and analysis.
Author Response
Dear Reviewer,
Thank you very much for taking the time to review the article Bioactive Compound Production of Grammatophyllum speciosum Blume Using Bioreactor Under Elicitation with Sodium Chloride.
Thank you again for your time and consideration
Sincerely,
Authors